# The Usefulness of Anthropometric Indices to Identify the Risk of Metabolic Syndrome

**DOI:** 10.3390/nu11112598

**Published:** 2019-10-29

**Authors:** Edyta Suliga, Elzbieta Ciesla, Martyna Głuszek-Osuch, Tomasz Rogula, Stanisław Głuszek, Dorota Kozieł

**Affiliations:** 1The Institute of Health Sciences, Medical College, Jan Kochanowski University, ul. Zeromskiego 5, 25-369 Kielce, Poland; eciesla@ujk.edu.pl (E.C.); mgluszekosuch@ujk.edu.pl (M.G.-O.); dorota.koziel@ujk.edu.pl (D.K.); 2The Institute of Medical Sciences, Medical College, Jan Kochanowski University, ul. Zeromskiego 5, 25-369 Kielce, Poland; tomrogula@gmail.com (T.R.); sgluszek@wp.pl (S.G.)

**Keywords:** a body shape index (ABSI), body roundness index (BRI), Clínica Universidad de Navarra-body adiposity estimator (CUN-BAE), body mass index (BMI), waist-to-height ratio (WHtR), percent of body fat (%BF), metabolic syndrome

## Abstract

Despite several papers having been published on the association between adiposity and the risk of metabolic syndrome (MetS), it is still difficult to determine unambiguously which of the indices of nutritional status is the best to identify MetS. The aim of this study was to analyze the ability of six anthropometric indices to identify MetS in the Polish population. The highest odds ratios for the occurrence of MetS, according to International Diabetes Federation (IDF), were noted for the following indices: waist-to-height ratio (WHtR, OR = 24.87) and Clínica Universidad de Navarra-body adiposity estimator (CUN-BAE, OR = 17.47) in men and WHtR (OR = 25.61) and body roundness index (BRI, OR = 16.44) in women. The highest odds ratios for the modified definition of MetS (without waist circumference) were found for the following indices: WHtR (OR = 7.32), BRI (OR = 6.57), and CUN-BAE (OR = 6.12) in women and CUN-BAE (OR = 5.83), WHtR (OR = 5.70), and body mass index (BMI, OR = 5.65) in men (*p* < 0.001 for all). According to the Receiver Operating Characteristic (ROC) analyses conducted for the identification of MetS, defined in accordance with IDF, the largest areas under the curve (AUCs) in men were observed for WHtR and CUN-BAE indices, whereas in women, they were observed for WHtR and BRI. In the analysis carried out for the identification of MetS (according to modified definition, without waist circumference), the AUCs were larger for WHtR and BRI in women, while in men, they were larger for CUN-BAE, BMI, and WHtR. BMI was also characterized by a relatively strong discriminatory power in identifying individuals with MetS. An optimal cut-off point for MetS, in accordance with the conventional definition, for both sexes was the value of BMI = 27.2 kg/m^2^. The weakest predictor of the syndrome was the ABSI (a body shape index) indicator. The most useful anthropometric indicator for the identification of MetS, both in men and in women in the Polish population, was WHtR. The optimal cut-off points for WHtR equaled 0.56 in men and 0.54 in women.

## 1. Introduction

Obesity is a significant risk factor for the development of several diseases, such as type 2 diabetes [1], metabolic syndrome (MetS) [2,3], cardiovascular diseases [4,5], numerous tumors [6,7,8], and musculoskeletal disorders [5], as well as a cause of high mortality [8,9]. BMI is most often used to evaluate obesity [10,11]. This indicator is simple, easy to calculate, and has clearly defined cut-off points. Because of its low cost, it is used in research worldwide and it enables the comparison of nutritional statuses in different populations. Limitations of BMI are its low accuracy in the assessment of adipose tissue and the fact that it does not allow for sex dimorphism and ethnic differences in adiposity, adipose tissue distribution, and age-related body composition [12,13,14]. Despite these limitations, the level of obesity determined on the basis of BMI is one of the main criteria for the application of pharmacological and surgical intervention [15]. Waist-to-height ratio (WHtR) and percent body fat (%BF) are relatively frequently used adiposity measures. However, continuous attempts are being made to devise new anthropometric measures of nutritional status, which would better identify a higher risk of morbidity and mortality. Relatively recently constructed indicators involve, among others, the following: a body shape index (ABSI) [16], body roundness index (BRI) [17], and Clínica Universidad de Navarra-body adiposity estimator (CUN-BAE) [18]. ABSI was devised in such a manner to be minimally correlated with height, mass, and BMI [16]. A high ABSI value indicates the waist circumference is higher than expected for a given height and mass and is related to a more central concentration of body volume. Body shape, measured with ABSI, seems to be an important risk factor of premature mortality in the general population. It can be used along with BMI to differentiate independent contributions of waist circumference and BMI to cardio-metabolic outcomes. BRI enables one to determine the shape of the human figure as an ellipse, generated from the height and waist circumference. BRI values are in the range from 1 to 16, and individuals with a more rounded figure are characterized by greater BRI values. It is a predictor of the percentage of adipose tissue and visceral tissue and can be a useful tool in the assessment of the health status [17]. The CUN-BAE indicator was suggested for the evaluation of the percentage of fat content in the body and it is calculated using BMI, sex, and age. Fat percentage calculated by means of CUN-BAE showed a strong correlation with the real content of adipose tissue (r = 0.89) [18]. In the authors’ opinion, this indicator can constitute an effective tool for identifying individuals at risk for Type 2 diabetes and cardiovascular diseases. 

MetS is defined as the accumulation of risk factors such as abdominal obesity, elevated blood pressure, dyslipidemia, and abnormal glycemia. It poses one of the main challenges for global and national public health institutions, as it is associated with a significantly higher risk of type 2 diabetes, cardiovascular diseases, some cancers, and all-cause mortality. Despite the fact that several papers have been published on the association between adiposity and the risk of MetS, it is still difficult to determine unambiguously which indicator of nutritional status is the best tool in order to identify individuals with MetS. To our knowledge, so far, there has been only one such study conducted in Poland, which involved the analysis of the relationships between ABSI and metabolic risk factors in a small group of 114 young men [19], whereas BRI and CUN-BAE have not yet been used for the assessment of health status and metabolic risk in our population. Hence, the aim of this study was to analyze the predictive capacity of new anthropometric indices, such as the ABSI, BRI, and CUN-BAE, in order to identify MetS in the Polish population and compare their usefulness to that of traditionally applied measures such as BMI, WHtR, and %BF.

## 2. Subjects and Methods

### 2.1. Study Design and Sample Collection

The basis of our study involved data obtained from the POlish-Norwegian Study (PONS) project [20,21]. It was a cross-sectional study on the health status of the adult Polish population from the region of Swietokrzyskie, conducted in the years 2010–2012. The study included analyses of collected fasting blood samples, blood pressure, and anthropometric measurements. Extensive interviews were carried out in order to collect information about the participants’ lifestyles during the last year. 13,172 participants, aged 37–66, volunteered to take part in the study. In further analyses, the data of 12,328 individuals were used (8234 women), and 844 participants were rejected due to incomplete data.

### 2.2. Ethical Approval

The study was approved by the Ethics Committee from the Cancer Centre and Institute of Oncology in Warsaw, No. 69/2009/1/2011 (data collection), and by the Committee on Bioethics at the Faculty of Health Sciences, Jan Kochanowski University in Kielce, Poland, No. 45/2016 (data analysis).

### 2.3. Measurements and Anthropometric Indicators 

Height was measured without shoes to the nearest 0.1 cm using a SECA stadiometer. Non-elastic tape was used to measure waist circumference (WC) at a point midway between the lowest rib and the iliac crest. Weight and %BF were measured using a body composition analyzer (Tanita SC 240MA, Tanita Corp., Tokyo, Japan) with an accuracy of 0.1 kg and 0.1%. On the basis of the performed measurements, the following indicators were calculated:

BMI = Weight (kg)/Height (m)^2^;WHtR = WC (cm)/Height (cm);ABSI = WC (m)/[BMI^2/3^(kg/m^2^) Height^1/2^ (m)] [16];$\mathrm{BRI}=364.2-365.5\times \sqrt{1-\left[\frac{{\left(WC/\left(2\pi \right)\right)}^{2}}{{\left(0.5\;\times \;Height\right)}^{2}}\right]}$ [17];CUN-BAE was calculated using the equation %BF = − 44.988 + (0.503 × age) + (10.689 × sex) + (3.172 × BMI) − (0.026 × BMI^2^) + (0.181 × BMI × sex) − (0.02 × BMI × age) − (0.005 × BMI^2^ × sex) + (0.00021 × BMI^2^ × age), where age is measured in years, and sex was codified as 0 for men and 1 for women [18].

### 2.4. Blood Pressure and Blood Biochemical Parameters 

Blood pressure (BP) was measured using an Omron blood pressure monitor (Model M3 Intellisense, Mannheim, Germany). BP was measured on the artery of the right upper limb when the participant was seated. In the study, an average of two measurements were used for analysis. Fasting blood samples were obtained for measurements of glucose and lipids using standard techniques. Blood was centrifuged for 10 minutes at 3500 rpm at room temperature. The glucose concentration in the blood serum was measured using the enzyme method with hexokinase, while the concentration of triglycerides (TGs) was assessed using the phosphogliceride oxidaseperoxidase method. The concentration of HDL cholesterol was obtained using the colorimetric non-precipitation method. Laboratory tests were performed with Integra 800 (La Roche Diagnostics, Switzerland). 

### 2.5. Socio-Demographic and Lifestyle Data 

Socio-demographic data and information on the participants’ lifestyles were collected in face-to-face interviews using structured questionnaires. The socio-demographic variables included sex, age, and education (number of education years in total). As an indicator of socioeconomic position, we chose education, which, in our opinion, reflects health awareness and health behavior patterns better than income. Moreover, education is comparatively easy to measure in self-administered questionnaires, garners a high response rate, and is relevant to people regardless of age or working circumstances, unlike many other indicators of socioeconomic position. For each participant, the daily intake of pure ethanol was calculated, with the consideration of the mean alcohol content in specific alcoholic beverages and the frequency of intake (g/week). Smoking was evaluated on the basis of the analysis of the prevalence of current, former, and never smoking behaviors. The respondents who smoked cigarettes during the study were classified as current smokers, and those who had not smoked for 6 months were classified as former smokers. The rest of the participants composed the group of nonsmokers. Physical activity (PA) was evaluated with the use of the International Physical Activity Questionnaire (IPAQ), long form. Total PA was calculated and expressed as metabolic equivalents (MET/min/week^−1^) [22]. 

### 2.6. The Definition of Metabolic Syndrome (MetS)

Following the criteria established by the International Diabetes Federation Task Force on Epidemiology and Prevention (joint interim statement in 2009) [23], MetS was defined as the presence of three or more of the following five components: abdominal obesity, WC ≥ 94 cm in males and ≥80 cm in females; fasting glucose ≥100 mg/dl (5.5 mmol/L) or diabetes treatment; triglycerides ≥150 mg/dL (1.7 mmol/L) or drug treatment for elevated triglycerides; HDL cholesterol <40 mg/dL (1.0 mmol/L) in males and <50 mg/dl (1.3 mmol/L) in females or drug treatment for reduced HDL cholesterol; and systolic blood pressure ≥130 mmHg or diastolic blood pressure ≥85 mmHg or drug treatment for hypertension. Allowing for very strong correlations with one of the MetS components (i.e., WC) with all anthropometric indicators, analyses were performed twice: (1) for a standard definition of MetS, including three out of five components according to the International Diabetes Federation (IDF (IDF), and (2) for a modified definition of MetS, including two out of four components other than WC.

### 2.7. Statistical Analysis

All continuous variables were expressed as means and standard deviations (X ± SD) and medians (Me) and interquartile categories (Q1–Q3). All categorical variables were reported as frequency and percentage (N, %). Comparisons between the groups of men and women were conducted using U Mann–Whitney or Chi-squared tests depending on the distribution of each feature. Spearman’s rank correlation coefficients were calculated to test the association between anthropometric measures and indices. Multivariate logistic regression was used to evaluate the unadjusted and adjusted associations between anthropometric measures and MetS. Quintiles of anthropometric indices were created, and odds ratios (ORs) of MetS and 95% confidence intervals (CIs) were calculated in each quintile. For all six anthropometric indices, the lowest quintile was set as reference. Adjustments were made for age, education, physical activity, alcohol consumption (continuous variables), and smoking status (categorical variable). The analyses were performed separately for men and women. Receiver Operating Characteristic (ROC) analyses were used to compare the predictive ability and to determine the optimal cut-off values of the anthropometric indices. We estimated the area under the curve (AUC) with 95% CIs. The AUC represents a measure of accuracy of each anthropometric index to discriminate between subjects with or without MetS. The optimal cut-off point was the highest Youden index value (sensitivity + specificity − 1). A *p* value <0.05 was assumed statistically significant for all calculations. All data were analyzed using Statistical Package Statistica software (Tibco software version 13.1, Warsaw, PL, Poland). 

## 3. Results

The average age of the participants was 55.95 ± 5.43 years for men and 55.54 ± 5.35 years for women. In the subject group, there were 44.0% of overweight individuals (BMI = 25.0–29.9 kg/m^2^) and 30.0% of individuals with obesity (BMI ≥30.0 kg/m^2^). Men were characterized by higher values of all anthropometric features and indicators of nutritional status than women (Table 1). MetS according to IDF was found in 42.4% of all subjects (Table 2). It occurred significantly more often in males than in females (*p* < 0.001). Among MetS components, only a lower HDL cholesterol concentration was found in women compared to men. No significant differences of physical activity depending on sex were noted (Table 3). Men smoked significantly more often and consumed more alcohol than women. They were also characterized by a shorter period of education, compared to women.

For all subjects, CUN-BAE was the indicator most strongly and negatively correlated with height (r = −0.591; *p* < 0.001) (Appendix A). In the analyses performed separately in groups according to sex, WHtR and BRI were correlated most strongly with height, whereas ABSI was not significantly correlated with height, either in men or in women. BMI was most strongly correlated with mass in all subjects (r = 0.818; p < 0.001), whereas in the analyses performed separately for both sexes, BMI, CUN-BAE, and %BF were the most strongly correlated with mass. BMI was also most strongly correlated with WHtR (r = 0.877; *p* < 0.001) and BRI (r = 0.877; *p* < 0.001), whereas in the analysis performed separately for both sexes, BMI was most strongly correlated with CUN-BAE (Appendix A). Among other indicators, WHtR and BRI were the indicators most strongly correlated with each other, both in all subjects and in both gender groups (r = 0.999; *p* < 0.001). The correlation coefficient r between CUN-BAE and %BF equaled 0.873 for all subjects and was slightly higher in females (r = 0.888) compared to males (r = 0.808) (*p* < 0.001 for all). WHtR and BRI were the indicators most strongly correlated with waist circumference.

In unadjusted models, the odds ratios for MetS grew along with quartiles for all six anthropometric indicators (Appendix A). However, they were much higher for the standard definition of MetS, including three out of five components, than for the modified definition, allowing for two out of four components (except WC). The risk of MetS was not significantly higher only in the second quartile of ABSI indicator in males.

The odds ratios in models adjusted for age, level of education, smoking, alcohol consumption, and physical activity also grew along with quartiles for all analyzed anthropometric indices (Table 4). However, they were significantly lower compared to those in the unadjusted models. The highest odds ratios for the occurrence of MetS, in accordance with the standard definition, were noted for the following indicators: WHtR (OR = 24.87, 95%CI: 18.85–32.82; *p* < 0.001) and CUN-BAE (OR = 17.47, 95%CI: 13.51–22.59; *p* < 0.001) in men and WHtR (OR = 25.61, 95%CI: 20.26–202.40; *p* < 0.001) and BRI (OR = 16.44, 95%CI: 13.32–20.29; *p* < 0.001) in women. The odds ratios for the modified MetS definition, including two out of four components, were significantly lower. In the fifth quintile, compared to the first one, the highest odds ratios in women were found for WHtR (OR = 7.32, 95%CI: 6.16–8.69; *p* < 0.001), BRI (OR = 6.57, 95%CI: 5.53–7.80; *p* < 0.001), and CUN-BAE (OR = 6.12 (5.17–7.25; *p* < 0.001). In men, the odds ratios in the fifth quintile were lower than in females for every analyzed indicator. The relatively highest values were noted for CUN-BAE (OR = 5.83, 95%CI: 4.66–7.31; *p* < 0.001), WHtR (OR = 5.70, 95%CI: 4.55–7.14; *p* < 0.001), and BMI (OR = 5.65, 95%CI: 4.53–7.05; *p* < 0.001). The risk of MetS was not significantly higher only in the second and third quartiles of the ABSI indicator in males.

According to the ROC analyses conducted for the MetS definition according to the IDF, the largest AUCs in men were observed for the WHtR and CUN-BAE indicators (0.764 and 0.760, respectively), whereas in women, the largest AUCs were for WHtR (0.758) and BRI (0.748) (Table 5, Figure 1). The smallest AUC was for ABSI, equaling 0.609 in men and 0.639 in women. In the analysis carried out for the modified definition of MetS (without WC), the AUC was the largest for WHtR (0.706) and BRI (0.701) in females, whereas in males, the largest AUC was for CUN-BAE (0.682) (Table 5, Figure 2). The lowest AUC values, like in a previous analysis, were noted for ABSI (0.610 in females and 0.551 in males). In the standard definition, the highest values of the Youden index were obtained for CUN-BAE (0.41) and WHtR (0.40) in men and for WHtR (0.38) in women. In the analysis conducted for the modified definition of MetS, the lowest values of Youdena index were found for WHtR and BRI in women (both values = 0.30) and for CUN-BAE and BRI in men (both values = 0.28). The optimal cut-off values of the best adiposity indices for the conventional MetS definition were as follows: 0.556 for WHtR and 29.04 for CUN-BAE in men and 0.535 for WHtR and 5.05 for BRI in women. In the analysis involving the modified MetS definition, MetS cut-off points were as follows: 0.571 for WHtR and 29.99 for CUN-BAE in males and 0.543 for WHtR and 5.05 BRI in females. It should also be stated that the values of optimal cut-off points for particular indices in the case of both MetS definitions differed slightly or (as in the case of BRI in women) did not differ at all. Moreover, it should be stressed that for CUN-BAE, the optimal cut-off points were slightly higher than for %BF measured by the bioimpedance method.

## 4. Discussion

The comparison of the odds ratios of MetS occurrence between the highest quintile (Q5) and the lowest (Q1) of every anthropometric index showed that the best MetS predictors, according to the standard MetS definition, were WHtR and CUN-BAE in men and WHtR and BRI in women. In the modified definition (without WC), the best MetS predictor in males was CUN-BAE followed by WHtR and BMI, whereas the best predictor of MetS in females was still WHtR, followed by BRI and CUN-BAE. The results obtained by the logistic regression analysis were also confirmed by the ROC curve analysis, showing that WHtR and CUN-BAE in the case of men and WHtR and BRI in the case of women should be considered the best anthropometric indices to identify MetS in the Polish population. Among these three indices, WHtR has the advantages of simplicity, low cost, similar usefulness both in men and in women, and a well-proven applicability in various populations. Ashwell et al. [24], on the basis of a meta-analysis of studies conducted in different ethnic groups, confirmed that, both in males and in females, WHtR was a better screening tool than BMI and WC for adult cardiometabolic risk factors. Similar results were obtained in subsequent studies, which compared various anthropometric indices, including newly devised ones [25,26,27]. Cut-off points for WHtR in the subject population equaled 0.56 for men and 0.54 for women. Similar values were achieved in the Brazilian population (≥0.54 for men and ≥0.55 for women), in which 52% of the participants were white [28]. However, slightly higher cut-offs were suggested in Turkey (0.58 for men and 0.59 for women) [29] and in the USA (0.58 for men and women) [30]. In the previously mentioned study conducted in the USA, more than half of the participants were NonHispanic Caucasians.

The usefulness of four further indices (i.e., CUN-BAE, BRI, BMI, and %BF) for the identification of individuals with MetS in the Polish population was also satisfactory and just slightly lower than that of WHtR. However, different discriminatory powers of particular indices were found depending on sex. In the analyses conducted for the modified definition of MetS among men, the largest AUC in ROC analyses was for CUN-BAE. Gomes-Marcos et al. also showed that, in a global analysis, the adiposity index that showed the highest odds ratio of MetS was CUN-BAE [31]. The results of our studies also revealed that the CUN-BAE index can relatively precisely assess the amount of adipose tissue. The correlation of CUN-BAE with %BF, measured by the bioimpedance method, was r = 0.873. A similar correlation of CUN-BAE with the actual amount of adipose tissue, determined by air displacement plethysmography (r = 0.89), was found by the index authors in the Spanish population [18].

The results of the analyses performed by us also revealed the relatively high usefulness of BRI to identify individuals with MetS. It should be emphasized that this index has shown a relatively strong correlation with WHtR (r = 0.999) in the Dutch population [32]. Its ability to identify metabolic disorders in both sexes has been confirmed by several studies [25,33,34]. The usefulness of BRI in our population was, by far, greater for females than for males, as in the Spanish population [31]. 

The usefulness of BMI in the evaluation of the risk of MetS occurrence was not significantly different from the usefulness of other indices and was clearly higher than that of ABSI. An optimal cut-off point for MetS, based on the standard definition, in our population for both sexes was the value of BMI = 27.2 kg/m^2^. A similar cut-off point for MetS both in men and in women (BMI = 27 kg/m^2^) was obtained by Ofer et al. [35]. Despite the fact that several papers stress the limitations of BMI [12,13,14], the results of our study, as well as some papers of other authors [31,36], suggest that BMI can be equally and sometimes even more useful clinically than other measures of obesity, assessed by means of more precise and more expensive methods. 

The discriminatory power of %BF was slightly lower not only than those of WHtR, CUN-BAE, and BRI but also than that of BMI. It may result from the fact that in MetS development, the distribution of adipose tissue is more important than its percentage in the body, as underlined by several authors [37,38,39]. Moreover, it may also prove a small precision of the measurement of fat content by means of electric bioimpedance. However, even a more accurate measurement of total adipose tissue, i.e., the hydrostatic weighing method, defined as the gold standard in the study conducted by Ortega et al., did not show a greater usefulness of this index in the evaluation of cardiovascular disease mortality compared to BMI [36].

ABSI was an exception among the analyzed parameters because, despite the fact that it turned out to be significant, in all analyses it was definitely the weakest predictor of MetS. This result is in compliance with the findings of most other authors, who concluded in their studies that the lowest AUC for MetS and other cardiometabolic risk factors belonged to ABSI [25,31,34,40,41,42]. Only a few studies suggest that the combination of BMI and ABSI may be more effective than the standard usage of BMI or any other single index in clinical practice [43]. In men in the Spanish population, ABSI turned out to be a better anthropometric cardiovascular risk indicator than WHR and BMI [27]. Similarly, in the Chinese population, it was concluded that ABSI was the best anthropometric index for estimating the risk of coronary heart disease in males [44].

### 4.1. Limitations 

Several limitations of the present study should be considered. Firstly, it was a cross-sectional study; therefore, no conclusions can be drawn regarding the changes in the anthropometric measures over time. Secondly, the study was limited to participants coming from only one region of Poland; hence, the applicability of these results may be limited for other populations. It should be taken into account that, among representatives of different populations of Caucasian origin, there are differences in body height and proportions that can change the relationship between body fat indicators and MetS risk [45]. In addition, in each population, there may also be different intensities of various confounding factors that make up lifestyle, which can also modify the above associations. As an indicator of socioeconomic position, we chose education because of the high non-response rate for income data. In the analysis, we also could not include an important confounding factor, i.e., the caloric value of the diet. Thirdly, this study defined MetS using IDF 2009 criteria. Therefore, further studies are needed to determine whether the results are consistent under different criteria. The obtained results should be approached with great caution also because of the limitations associated with the design of the indicators themselves. Their predictive models were, in many cases, developed on small numbers and were also based on inaccurate body composition analysis techniques, such as skinfolds or bioelectrical impedance analyses. The lowest level of ABSI discriminatory power can be explained by the fact that this indicator was constructed to predict the risk of mortality in long-term studies, while we used it as a MetS predictor in our cross-sectional study.

### 4.2. Strengths 

An advantage of the study is, first and foremost, the large sample size. Moreover, the anthropometric indices of the participants were calculated on the basis of actual measurements and were not self-reported. Because the anthropometric cut-off points could be different for men and women, statistical analyses were performed separately for men and women. The analysis included numerous confounders related to MetS, such as physical activity, smoking, and alcohol consumption. 

## 5. Conclusions

An anthropometric index showing greatest usefulness in MetS identification in the Polish population is WHtR. Optimal cut-off points for WHtR equaled 0.56 in men and 0.54 in women. Indices recommended subsequently are CUN-BAE for men and BRI for women. The results of our study confirmed a satisfactory usefulness of BMI and %BF for identifying MetS, whereas ABSI was found to be the weakest predictor of the syndrome. Our results showed that in a population where the average BMI was 28 kg/m^2^, over 57% of the participants were healthy (without MetS). Therefore, it may suggest that in using the above indicators and the proposed cut-off points, a large percentage of individuals were diagnosed as false positive. However, studies with a follow-up period of at least 10 years showed that obese individuals who were metabolically healthy had an increased risk of cardiovascular events and all-cause mortality compared to metabolically healthy individuals with a normal body weight [46]. Therefore, the cut-offs proposed in this study provide an earlier diagnosis of MetS than the commonly accepted obesity criterion (i.e., BMI ≥30 kg/m^2^). In our analysis, we included the classic MetS definition (i.e., three of five components according to the IDF) and a modified definition (i.e., two of four components other than WC). To avoid late diagnosis of MetS, consideration should be given to setting cut-off points for the indicators in question that would allow people with only one MetS component to be diagnosed.

## Figures and Tables

**Figure 1 nutrients-11-02598-f001:**
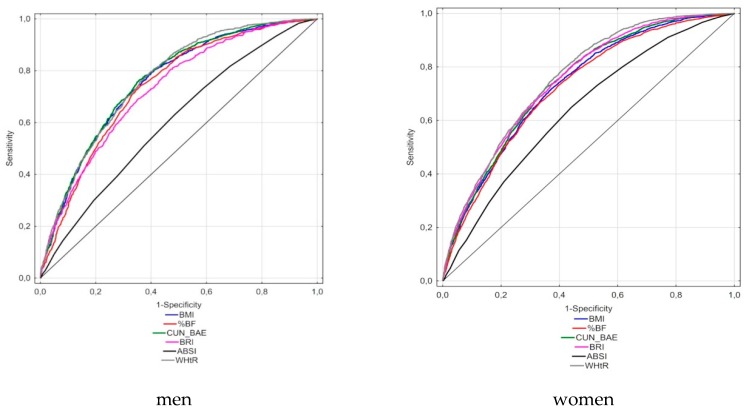
Receiver-operating characteristic curves of anthropometric indices to identify men and women with MetS (MetS classic: three or more components out of five).

**Figure 2 nutrients-11-02598-f002:**
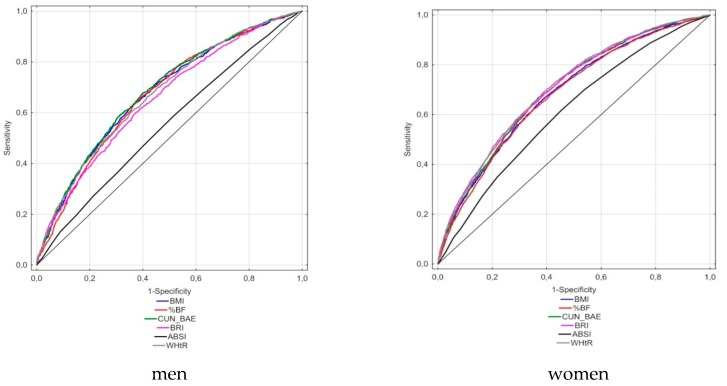
Receiver-operating characteristic curves of anthropometric indices to identify men and women with MetS (MetS modified: two or more components out of four, other than WC).

**Table 1 nutrients-11-02598-t001:** Anthropometric measurements and adiposity indexes of the subject groups.

Variables.	Total (N = 12328) X ± SD; Me (Q1–Q3)	Males (N = 4094) X ± SD; Me (Q1–Q3)	Females (N = 8234) X ± SD; Me (Q1–Q3)	*p* Value
Body height [cm]	164.35 (8.64); 163.00 (158.00–170.00)	173.25 (6.29); 173.00 (169.00–177.00)	159.93 (5.75); 160.00 (156.00–164.00)	<0.001
Body mass [kg]	76.16 (14.49); 74.70 (65.60–85.30)	85.50 (13.04); 84.70 (76.50–93.00)	71.52 (12.84); 70.00 (62.50–78.80)	<0.001
Waist circumference [cm]	91.80 (12.57); 91.00 (83.00–100.00)	99.18 (10.41); 99.00 (92.00–105.00)	88.12 (11.92); 87.00 (80.00–96.00)	<0.001
BMI [kg/m^2^]	28.15 (4.66); 27.63 (24.87–30.79)	28.46 (3.95); 28.17 (25.77–30.72)	27.99 (4.96); 27.29 (24.42–30.84)	<0.001
WHtR	55.88 (7.33); 55.45 (55.89–60.47)	57.30 (6.12); 57.06 (53.18–60.80)	55.18 (7.77); 54.43 (49.68–60.06)	<0.001
%BF	33.05 (7.85); 33.40 (27.30–38.90)	27.00 (6.47); 26.50 (22.70–30.90)	36.00 (6.72); 36.50 (31.70–40.70)	0.001
ABSI [m^11/6^ · kg^−2/3^]	0.077 (0.005); 0.078 (0.074–0.081)	0.081 (0.004); 0.081 (0.078–0.083)	0.076 (0.005); 0.076 (0.073–0.079)	<0.001
BRI	4.639 (1.597); 4.444 (3.529–5.540)	4.909 (1.362); 4.784 (3.980–5.658)	4.505 (1.686); 4.233 (3.303–5.449)	<0.001
CUN–BAE [%]	37.10 (7.42); 37.28 (31.62–42.45)	29.81 (4.87); 29.72 (26.59–32.82)	40.72 (5.58); 40.52 (36.72–44.47)	<0.001

N, number of participants; X ± SD, arithmetic mean ± standard deviation; Me, median; Q, quintile; BMI, body mass index; WHtR, waist-to-height ratio; %BF, percent of body Fat; ABSI, a body shape index; BRI, body roundness Index; CUN-BAE, Clínica Universidad de Navarra-body adiposity estimator.

**Table 2 nutrients-11-02598-t002:** Metabolic syndrome and its components in the subject groups.

Variables.		Total (N = 12328) N (%)	Males (N = 4094) N (%)	Females (N = 8234) N (%)	*p* Value
MetS	Yes	5227 (42.40)	2036 (49.73)	3191 (38.75)	<0.001
	No	7101 (57.60)	2058 (50.27)	5043 (61.25)	
Glucose	Yes	4109 (33.33)	1857 (43.36)	2252 (27.35)	<0.001
	No	8219 (66.67)	2237 (54.64)	5982 (72.65)	
Abdominal obesity	Yes	3219 (26.11)	1185 (28.94)	2034 (24.70)	<0.001
	No	9109 (73.89)	2909 (71.06)	6200 (75.30)	
HDL cholesterol	Yes	2239 (18.16)	670 (16.37)	1569 (19.06)	<0.001
	No	10089 (81.84)	3424 (83.63)	6665 (80.94)	
TG	Yes	4226 (34.28)	1642 (40.11)	2584 (31.38)	<0.001
	No	8102 (65.72)	2452 (59.89)	5650 (68.62)	
Elevated BP	Yes	9136 (74.11)	3372 (82.36)	5764 (70.00)	<0.001
	No	3192 (25.89)	722 (17.64)	2470 (30.00)	

N, number of participants; MetS, metabolic syndrome; High Density Lipoproteins, HDL, TG, triglycerides; BP, blood pressure.

**Table 3 nutrients-11-02598-t003:** Social variables and lifestyle habits of the subject groups.

Variables.	Total (N = 12328) X ± SD; Me (Q1–Q3)	Males (N = 4094) X ± SD; Me (Q1–Q3)	Females (N = 8234) X ± SD; Me (Q1–Q3)	*p*-Value
Years of education	13.23 (3.18); 13.00 (11.00–16.00)	13.22 (3.20); 12.00 (11.00–16.00)	13.24 (3.17); 13.00 (11.00–16.00)	0.024
Physical activity [METs/min/week^−1^]	4499.0 (3640.1); 3492.0 (1833.0–6180.0)	4636.2 (3954.1); 3600.0 (1674.0–6675.0)	4430.8 (3471.7); 3446.3 (1890.0–5970.0)	0.889
Alcohol [g/week]	40.40 (95.07); 13.44 (2.80–39.53)	86.54 (146.91); 45.76 (17.26–100.7)	17.47 (34.91); 7.53 (1.84–19.84)	<0.001
Nonsmokers	N = 5781 (46.89%)	N = 1452 (35.47%)	N = 4329 (52.27%)	<0.001
Former smokers	N = 4152 (33.68%)	N = 1758 (42.94%)	N = 2394 (29.07%)	
Current smokers	N = 2395 (19.43%)	N = 884 (21.59%)	N = 1511 (18.35%)	

N, number of participants; X ± SD, arithmetic mean ± standard deviation; Me, median; Q, quintile; MET, metabolic equivalent.

**Table 4 nutrients-11-02598-t004:** Odds ratios and 95% confidence intervals for MetS, adjusted for age, education, smoking, physical activity, and alcohol consumption.

Indices.	Q	Men				Women			
		MetS Classic Definition (3 or More Components Out of 5)		MetS Modified Definition (2 or More Components Out of 4, Other than WC)		MetS Classic Definition (3 or More Components Out of 5)		MetS Modified Definition (2 or More Components Out of 4, Other than WC)	
		OR (95% CI)	*p*	OR (95% CI)	*p*	OR (95% CI)	*p*	OR (95% CI)	*p*
BMI	1(ref.)	1.0		1.0		1.0		1.0	
	2	3.05 (2.40–3.89)	<0.001	1.60 (1.31–1.96)	<0.001	2.94 (2.42–3.56)	<0.001	1.70 (1.44–2.00)	<0.001
	3	6.63 (5.21–8.44)	<0.001	2.51 (2.04–3.08)	<0.001	4.92 (4.08–5.95)	<0.001	2.44 (2.07–2.86)	<0.001
	4	11.02 (8.62–14.09)	<0.001	3.72 (3.01–4.59)	<0.001	7.82 (6.47–9.44)	<0.001	3.61 (3.07–4.24)	<0.001
	5	17.52 (13.57–22.62)	<0.001	5.65 (4.53–7.05)	<0.001	12.76 (10.53–15.46)	<0.001	5.90 (5.01–6.96)	<0.001
WHtR	1(ref.)	1.0		1.0		1.0		1.0	
	2	4.66 (3.59–6.05)	<0.001	1.77 (1.44–2.16)	<0.001	5.47 (4.33–6.90)	<0.001	1.91 (1.61–2.26)	<0.001
	3	9.25 (7.13–12.00)	<0.001	2.33 (1.89–2.85)	<0.001	10.11 (8.04–12.71)	<0.001	2.91 (2.47–3.44)	<0.001
	4	14.10 (10.83–18.37)	<0.001	3.28 (2.66–4.05)	<0.001	14.59 (11.60–18.35)	<0.001	4.19 (3.55–4.94)	<0.001
	5	24.87 (18.85–32.82)	<0.001	5.70 (4.55–7.14)	<0.001	25.61 (20.26–202.4)	<0.001	7.32 (6.16–8.69)	<0.001
%BF	1(ref.)	1.0		1.0		1.0		1.0	
	2	2.41 (1.91–3.04)	<0.001	1.60 (1.31–1.96)	<0.001	2.59 (2.14–3.12)	<0.001	1.77 (1.51–2.09)	<0.001
	3	5.00 (3.97–6.30)	<0.001	2.48 (2.02–3.05)	<0.001	4.30 (3.58–5.16)	<0.001	2.47 (2.11–2.90)	<0.001
	4	8.25 (6.52–10.43)	<0.001	3.49 (2.83–4.31)	<0.001	6.99 (5.83–8.39)	<0.001	3.79 (3.23–4.45)	<0.001
	5	11.29 (8.87–14.38)	<0.001	4.77 (3.84–5.94)	<0.001	9.96 (8.29–11.96)	<0.001	5.41 (4.60–6.36)	<0.001
ABSI	1(ref.)	1.0		1.0		1.0		1.0	
	2	1.58 (1.29–1.93)	<0.001	1.09 (0.90–1.33)	0.377	1.68 (1.43–1.98)	<0.001	1.39 (1.19–1.62)	<0.001
	3	1.85 (1.51–2.27)	<0.001	1.14 (0.93–1.39)	0.208	2.30 (1.96–2.70)	<0.001	1.80 (1.55–2.09)	<0.001
	4	2.03 (1.66–2.49)	<0.001	1.23 (1.00–1.50)	0.048	2.85 (2.43–3.34)	<0.001	2.08 (1.79–2.42)	<0.001
	5	2.46 (1.99–3.04)	<0.001	1.31 (1.06–1.61)	0.011	3.45 (2.94–4.05)	<0.001	2.44 (2.10–2.85)	<0.001
BRI	1(ref.)	1.0		1.0		1.0		1.0	
	2	2.49 (1.99–3.13)	<0.001	1.50 (1.23–1.84)	<0.001	3.63 (2.95–4.47)	<0.001	1.80 (1.53–2.13)	<0.001
	3	4.42 (3.52–5.53)	<0.001	2.07 (1.69–2.54)	<0.001	6.32 (5.15–7.75)	<0.001	2.71 (2.30–3.19)	<0.001
	4	6.68 (5.30–8.41)	<0.001	2.72 (2.20–3.35)	<0.001	9.55 (7.79–11.72)	<0.001	3.84 (3.25–4.52)	<0.001
	5	11.81 (9.25–15.09)	<0.001	4.40 (3.52–5.50)	<0.001	16.44 (13.32–20.29)	<0.001	6.57 (5.53–7.80)	<0.001
CUN–BAE	1(ref.)	1.0		1.0		1.0		1.0	
	2	2.85 (2.24–3.64)	<0.001	1.57 (1.29–1.93)	<0.001	2.91 (2.38–3.56)	<0.001	1.66 (1.41–1.97)	<0.001
	3	6.21 (4.89–7.87)	<0.001	2.46 (2.01–3.03)	<0.001	5.78 (4.75–7.03)	<0.001	2.74 (2.33–3.23)	<0.001
	4	9.92 (7.77–12.67)	<0.001	3.44 (2.79–4.26)	<0.001	8.48 (6.96–10.34)	<0.001	3.76 (3.19–4.44)	<0.001
	5	17.47 (13.51–22.59)	<0.001	5.83 (4.66–7.31)	<0.001	13.80 (11.28–16.87)	<0.001	6.12 (5.17–7.25)	<0.001

OR, odds ratio; CI, confidence interval; Q, quintile; WC, waist circumference; ref., reference level; BMI, Body Mass Index; WHtR, Waist-to-Height Ratio; %BF, Percent of Body Fat; ABSI, A Body Shape Index; BRI, Body Roundness Index; CUN-BAE, Clínica Universidad de Navarra-Body Adiposity Estimator.

**Table 5 nutrients-11-02598-t005:** The AUCs of each anthropometric index for the presence of MetS in both genders.

Indices	Gender	AUC	95%CI	*p*	Youden Index	Cut–Off Points
MetS classic (3 or more components out of 5)						
BMI	Men	0.754	0.739–0.769	<0.001	0.39	27.18
	Women	0.731	0.720–0.742	<0.001	0.35	27.20
WHtR	Men	0.764	0.749–0.778	<0.001	0.40	0.556
	Women	0.758	0.748–0.768	<0.001	0.38	0.535
%BF	Men	0.738	0.722–0.753	<0.001	0.38	25.80
	Women	0.722	0.711–0.733	<0.001	0.33	36.14
ABSI	Men	0.603	0.586–0.620	<0.001	0.15	0.081
	Women	0.639	0.627–0.651	<0.001	0.21	0.076
BRI	Men	0.728	0.713–0.743	<0.001	0.34	4.82
	Women	0.748	0.737–0.758	<0.001	0.36	5.05
CUN BAE	Men	0.760	0.746–0.775	<0.001	0.41	29.04
	Women	0.742	0.732–0.753	<0.001	0.37	40.54
MetS modified (2 or more components out of 4, other than WC)						
BMI	Men	0.675	0.659–0.692	<0.001	0.27	28.07
	Women	0.685	0.673–0.696	<0.001	0.28	27.46
WHtR	Men	0.672	0.656–0.689	<0.001	0.25	0.571
	Women	0.706	0.695–0.718	<0.001	0.30	0.543
%BF	Men	0.670	0.653–0.687	<0.001	0.27	25.80
	Women	0.679	0.667–0.691	<0.001	0.27	37.10
ABSI	Men	0.551	0.533–0.568	<0.001	0.07	0.081
	Women	0.610	0.598–0.622	<0.001	0.16	0.076
BRI	Men	0.654	0.637–0.671	<0.001	0.28	4.85
	Women	0.701	0.690–0.712	<0.001	0.29	5.05
CUN–BAE	Men	0.682	0.666–0.699	<0.001	0.28	29.99
	Women	0.679	0.667–0.708	<0.001	0.30	40.62

AUC, area under the curve.

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
