# Peer review of "The Usefulness of Anthropometric Indices to Identify the Risk of Metabolic Syndrome"

_nutrients, 2019, doi:10.3390/nu11112598_

Round 1

Reviewer 1 Report

This is an interesting paper on the usefulness of anthropometric indices to identify the risk of metabolic syndrome. The review on the various indices is written with many details allowing the understanding of differences between the indices. I suggest that the authors introduce briefly the criteria of the metabolic syndrome in the introduction because the important point is the risk of metabolic syndrome. Indices are tools. Another interesting point is the number of patients in the cohort which validate by itself the results.

May be I misunderstood a point but line 162 it is written that MetS occured significantly more often in females than in males. But in Table 2, for a BMI of 28, 38.75% of females versus 49.71% of males had MetS.

It is interesting to have discriminated between former and current smokers. However, i did not find anything special for this parameter throughout the text. The point is not even discussed. So what was the purpose?

Line 252: Polish population is specified. Could the authors specify what characteristics the Polish population may have versus another Caucasian population? could it be related to height? The authors also give values for the Brazilian population or the US population but they should be more precise as these populations are extremely heterogeneous. 

The authors also discuss which indice fits better to populations. From a mathematical point of view, they may be right but for medicine this is a peculiar way of reasoning. In addition, while the cohort has a BMI in the range of 28, 49% of men and 39% of women had developed MetS which means that more than 50% of the population with a BMI of 28 was metabolically "healthy". I suggest the authors include a paragraph highlighting the limits of the indices and what could be done to give an early diagnostic before disease development.

Author Response

Response to the review 1

I suggest that the authors introduce briefly the criteria of the metabolic syndrome in the introduction because the important point is the risk of metabolic syndrome. Indices are tools.

Criteria for metabolic syndrome are included in the introduction

Maybe I misunderstood a point but line 162 it is written that MetS occured significantly more often in females than in males. But in Table 2, for a BMI of 28, 38.75% of females versus 49.71% of males had MetS.

It was indeed a mistake that was corrected in the text. It should be as follows: metabolic syndrome occurred significantly more often in males than in females.

It is interesting to have discriminated between former and current smokers. However, I did not find anything special for this parameter throughout the text. The point is not even discussed. So, what was the purpose?

The distinction between current and former smokers is used in the works of many authors [including Gepner et al. Am Heart J 2011; 161 (1): 145-151 .; Aldaham et al. Int J Inflam 2015; 2015: 439396], because smoking cessation may be associated with an improvement in lipid profile compared to current smokers. However, due to the frequent increase in body fat after smoking cessation, the lipid profile may be worse than in those who have never smoked. This distinction was used because in our previous studies in this population, it turned out that being a current or former smoker may have a different effect on the risk of metabolic syndrome (MetS) [e.g. Suliga et al. PLoS ONE 2016; 11 (4): e0154511.]. The aim of this study was to examine the ability of anthropometric indices to identify MetS, and not to analyze lifestyle components affecting MetS, so we did not analyze these factors in the manuscript. They were used only as confounding factors in multifactorial logistic regression analysis.

Line 252: Polish population is specified. Could the authors specify what characteristics the Polish population may have versus another Caucasian population? could it be related to height? The authors also give values for the Brazilian population or the US population but they should be more precise as these populations are extremely heterogeneous.

We included relevant explanations in the manuscript. More precise information was also provided on the Brazilian and American populations.

The authors also discuss which indice fits better to populations. From a mathematical point of view, they may be right but for medicine this is a peculiar way of reasoning. In addition, while the cohort has a BMI in the range of 28, 49% of men and 39% of women had developed MetS which means that more than 50% of the population with a BMI of 28 was metabolically "healthy". I suggest the authors include a paragraph highlighting the limits of the indices and what could be done to give an early diagnostic before disease development.

As suggested by the reviewer, the manuscript provides information on limitations of indicators and an explanation of what should be done to diagnose the disease in advance.

Reviewer 2 Report

Overall Summary

The study by Suliga and colleagues entitled, “The usefulness of anthropometric indices to identify the risk of metabolic syndrome” aimed to compare various anthropometric indices and the association with MetS. Considering body fatness is highly associated with MetS, this study addresses a key issue that will help inform Polish community (and potentially other communities) of the best anthropometric assessment for predicting MetS. This study concluded that independent of sex, waist-to-height ratio may be the best predictor of MetS in the Polish population.

Major Concerns

-Section 2.5 considers socio-demographic and lifestyle data, but does not describe socio-economic status or dietary intake, which may be confounding variables for the data presented in this manuscript. Did authors consider assessing socio-economic status or normal dietary intake as co-variates in this study. If not, this should be explained and included in the limitations section.

-Graphs presented in Figures 1 & 2 should have better resolution.

Minor Concerns

-Should state centrifuge conditions in section 2.4

-Authors should discuss how these data may be translated to other countries rather than just stating it may be a limitation that this study was conducted solely in a Polish cohort

-Authors should read through the manuscript to look for any typos

Author Response

 Response to the review 2 

Major Concerns

-Section 2.5 considers socio-demographic and lifestyle data, but does not describe socio-economic status or dietary intake, which may be confounding variables for the data presented in this manuscript. Did authors consider assessing socio-economic status or normal dietary intake as co-variates in this study. If not, this should be explained and included in the limitations section.

We chose education as an indicator of socioeconomic position, (the number of years of education), which in our opinion reflects the level of health awareness and health behavior patterns better than income. In addition, education is comparatively easy to measure in self-administered questionnaires, garners a high response rate, and is relevant to people regardless of age or working circumstances, unlike many other indicators of socioeconomic position.

The Food Frequency Questionnaire used in the assessment of diet did not allow calculating the energy value of the diets of the examined people, therefore nutrition data were not included in the confounding factors.

-Graphs presented in Figures 1 & 2 should have better resolution.

The resolution of the graphs has been improved

Minor Concerns

-Should state centrifuge conditions in section 2.4

This information is included in section 2.4

-Authors should discuss how these data may be translated to other countries rather than just stating it may be a limitation that this study was conducted solely in a Polish cohort

The manuscript provides a brief discussion on the possibilities of using the results of the study in other Caucasian populations.

-Authors should read through the manuscript to look for any typos

We used the  MDPI’s English Editing Service

Round 2

Reviewer 1 Report

I have a last comment: It is not clear that the sentence line 278 in red refers to [30]

Author Response

Response to the reviewer

Comments and Suggestions for Authors

I have a last comment: It is not clear that the sentence line 278 in red refers to [30]

The sentence has been corrected as suggested by the reviewer.
